# Prevalence and Factors of Osteoporosis and High Risk of Osteoporotic Fracture in Patients with Ankylosing Spondylitis: A Multicenter Comparative Study of Bone Mineral Density and the Fracture Risk Assessment Tool

**DOI:** 10.3390/jcm11102830

**Published:** 2022-05-17

**Authors:** Ji-Won Kim, Sunghoon Park, Ju-Yang Jung, Hyoun-Ah Kim, Seong-Ryul Kwon, Sang Tae Choi, Sung-Soo Kim, Sang-Hyeon Kim, Chang-Hee Suh

**Affiliations:** 1Department of Rheumatology, Ajou University School of Medicine, Suwon 16499, Korea; jwk722@naver.com (J.-W.K.); serinne20@hanmail.net (J.-Y.J.); nakhada@naver.com (H.-A.K.); 2Department of Radiology, Ajou University School of Medicine, Suwon 16499, Korea; mmedpark@ajou.ac.kr; 3Division of Rheumatology, Department of Internal Medicine, InHa University College of Medicine, Incheon 22332, Korea; rhksr@inha.ac.kr; 4Division of Rheumatology, Department of Internal Medicine, Chung-Ang University College of Medicine, Seoul 06973, Korea; beconst@cau.ac.kr; 5Division of Rheumatology, Department of Internal Medicine, Kangneung Asan Medical Center, Ulsan University College of Medicine, Kangneung 25440, Korea; drkiss@ulsan.ac.kr; 6Division of Rheumatology, Department of Internal Medicine, Keimyung University College of Medicine, Daegu 41931, Korea; mdkim9111@dsmc.or.kr; 7Department of Molecular Science and Technology, Ajou University, Suwon 16499, Korea

**Keywords:** ankylosing spondylitis, osteoporosis, osteoporotic fracture, bone mineral density, fracture risk assessment tool

## Abstract

Background: We investigated the prevalence of and the factors associated with a high risk of osteoporotic fractures in Korean patients with ankylosing spondylitis (AS). Methods: This was a multicenter, retrospective study including 219 AS patients from five university hospitals; the control group was selected by matching age and sex with those of the AS patients. The fracture risk was evaluated based on bone mineral density (BMD) measured by dual-energy X-ray absorptiometry and the fracture risk assessment tool (FRAX) with/without BMD. Results: The mean age of the patients was 47.6 years, and 144 (65.8%) patients were men. According to the WHO criteria and FRAX with/without BMD, the candidates for pharmacological treatment were 44 (20.1%), 20 (13.2%), and 23 (15.1%) patients, respectively, significantly more than those in the healthy control group. Among them, the proportion of patients receiving osteoporosis treatment was 39.1–75%. In logistic regression analysis, menopause was an independent factor for the high risk of fracture according to the WHO criteria and FRAX with/without BMD. C-reactive protein level (odds ratio (OR) 3.8 and OR 6) and glucocorticoid use (OR 1.5 and OR 1.7) were associated with a high risk of osteoporotic fracture based on FRAX without BMD and osteoporosis diagnosed according to the WHO criteria. Conclusions: Our study suggests that both FRAX and WHO criteria may be complementary for treatment decisions to reduce osteoporotic fractures in patients with AS.

## 1. Introduction

Ankylosing spondylitis (AS), a type of spondyloarthritis (SpA), is a chronic and progressive inflammatory disease that mainly affects the spinal and sacroiliac joints [1,2]. The primary pathological process of SpA, enthesitis, is the inflammation of the ligaments and tendons attached to the bones, leading to fibrous tissue proliferation during cartilage and bone formation [2]. This process further produces bony spurs, syndesmophytes, and enthesophytes, which eventually leads to ankylosis, resulting in the loss of spinal mobility and spinal stiffness [3]. Unlike the normal spine, which prefers flexibility over rigidity, the spine of patients with AS behaves like a long bone because of its rigidity, which makes it increasingly vulnerable and unable to absorb even minimal impacts [4]. Therefore, AS is closely related to an increased risk of fracture, and the rate of vertebral fractures in patients with AS is 4–40%; this rate is reported to be more than four times higher than that in the general population [5,6]. Furthermore, vertebral fractures in AS are often unstable due to the ossification of the supportive and elastic soft tissues. Neurological deficits can occur due to dislocation, resulting in a much higher mortality rate in this patients than in the general population [7].

Along with rigidity, an important factor influencing the increased risk of vertebral fractures is low bone mineral density (BMD) [8]. Inflammatory cytokines including tumor necrosis factor (TNF)-alpha, interleukin (IL)-1, IL-6, and IL-17 (factors characteristically produced in AS) activate osteoclast-mediated bone resorption, resulting in bone destruction [9]. In addition, syndesmophytes and ankylosis increase the load in the cortical zones and paravertebral ligaments, and the loss of trabecular bone in the vertebral bodies may lead to low BMD [10]. BMD reduction proceeds through this process and occurs mainly within the first 10 years of the disease [11]. Since most patients are diagnosed with AS between the ages of 20 and 40 years, the evaluation of BMD is often overlooked because of their young age, despite the high risk of bone loss in the early stages of the disease. Paradoxically, in the elderly population of AS, BMD may be artificially increased during dual-energy X-ray absorptiometry (DXA) due to bone sclerosis in disease evolution [12,13]. Spinal DXA, especially in the anteroposterior view, is unreliable in that it may provide falsely elevated results due to syndesmophytes and ligament ossification [14]. The inaccurate measurements of spinal DXA overestimate BMD, and this misinterpretation can lead to fractures even when osteoporosis is not diagnosed. For this reason, alternative methods such as lateral lumbar BMD and volumetric BMD estimated by DXA, trabecular bone score, and quantitative computed tomography are being introduced; however, these methods are not widely available [15].

The combined burden of low BMD and extensive syndesmophyte formation poses a high risk of vertebral fractures in patients with AS [16]. Considering the fracture complications and mortality rates of patients with AS, it is recommended to prevent low BMD and fractures through nonpharmacological and pharmacological treatments [17,18]. The most commonly used osteoporosis criterion for determining a pharmacological treatment is the evaluation of BMD measured using DXA based on the World Health Organization (WHO) criteria [19]. However, patients with AS may not receive appropriate treatment because of an overestimation of their BMD; thus, alternative evaluations of the fracture risk may be necessary. The Fracture Risk Assessment Tool (FRAX), which is used to detect the risk of osteoporotic fractures, is considered an appropriate tool for determining pharmacological treatments. FRAX has been developed as an online tool to easily provide estimates of 10-year fracture probabilities with or without BMD determination. It is increasingly used in clinical practice to help decision making regarding fracture prevention and has the advantages of being able to consider various factors such as past history and comorbidities [20]. Therefore, in this study, among patients with ankylosing spondylitis, patients with osteoporosis according to the WHO criteria and those with a high risk of osteoporotic fractures according to FRAX calculation were identified. We also investigated the proportion of patients who were receiving treatment and examined the factors affecting osteoporotic fractures.

## 2. Materials and Methods

### 2.1. Study Population and Clinical Assessments

We retrospectively reviewed the records of patients with AS from January 2012 to June 2020 at five university hospitals in South Korea (Ajou University Hospital, Chung-Ang University Hospital, Inha University Hospital, Ulsan University Gangneung Asan Hospital, and Keimyung University Hospital). Institutional review board approval was obtained at all study centers, and a waiver of informed consent was permitted due to the retrospective nature of the study and lack of personal data (AJIRB-MED-MDB-15-285, C2015163 [1621], 2015-09-026, 3-32100191-AB-N-01, and DSMC2015-12-017-007). Inclusion criteria were patients aged ≥18 years who met the 1984 modified New York criteria for AS [21]. Patients with BMD measured using DXA and those who underwent thoracolumbar spine radiography were enrolled in this study. Patients with chronic liver or renal failure, malignancy, and other rheumatic diseases were excluded as these diseases can influence functional abilities and disease activities in AS. The patients’ medical records were reviewed to obtain data on demographic, clinical, and medical characteristics. Demographic data included information on age, sex, menopausal status, body mass index (BMI), alcohol consumption, and smoking habits. We collected laboratory findings such as erythrocyte sedimentation rate (ESR), C-reactive protein (CRP) level, human leukocyte antigen-B27 (HLA-B27) level, and medication histories such as treatment for AS and osteoporosis at the time of BMD measurement. The healthy controls (HCs) comprised age- and sex-matched individuals whose BMD was measured in regular health check-ups during the same study period and did not have any comorbidities. Medical records were reviewed for those who agreed to it.

### 2.2. Evaluation of Osteoporosis by the WHO Criteria and of Osteoporotic Fracture Risk by the FRAX

BMD was measured in the lumbar spine (L1–L4) and proximal femur (femoral neck and total hip) using DXA. BMD measurements were performed using the manufacturer’s standard scanning software and positioning protocols. Patients were classified as normal (T-score ≥ −1), with osteopenia (T-score < −1 and > −2.5), and with osteoporosis (T-score ≤ −2.5) [19]. The FRAX calculations were performed using the Korean model (https://www.sheffield.ac.uk/FRAX/tool.aspx?country=25, accessed on 1 June 2021) and in relation to sex, age, bone mass index, a prior fragility fracture, a parental history of hip fracture, current tobacco smoking, long-term use of oral glucocorticoids, presence of rheumatoid arthritis (RA), other causes of secondary osteoporosis, and daily alcohol consumption of three or more units daily [20]. A high risk of osteoporotic fractures based on FRAX was defined as a 10-year probability ≥20% for major (clinical spine, shoulder, hip, or wrist) osteoporotic fracture (MOF) or ≥3% for hip fracture (HF) alone. We calculated both FRAX with BMD and FRAX without BMD, and the femoral neck BMD values were used for FRAX with BMD.

### 2.3. Radiological Assessments

Spinal radiographs were evaluated for vertebral fractures and spinal involvement of AS. The history of fractures was assessed using radiologic evaluation and patient self-report questionnaires, and radiographs of vertebral fractures were independently scored by two trained experts without information about patient characteristics. Fractures defined as fragility fractures were included, and unrelated to osteoporosis or pathological fractures caused by malignancy or infection were excluded from the analysis. Radiologists qualitatively analyzed spine radiographs to identify vertebral fractures. To assess AS-related spinal alterations, the modified Stoke Ankylosing Spondylitis Spinal Score (mSASSS) was used [22]. The mSASSS is based on lateral radiographic views of the spine as a well-validated scoring system to quantify chronic structural damage. The corresponding nominal scoring system was as follows: 0, no abnormality; 1, erosion, sclerosis or squaring; 2, syndesmophyte; and 3, bridging syndesmophytes (ankylosis). The sum of the scores was calculated for the upper and lower edges of the 24 vertebrae borders, with the total sum score ranging from 0 to 72. The scoring system includes the lower border of the 12th thoracic vertebra, the upper/lower border of all five lumbar vertebrae, and the upper border of the sacrum as seen in the lateral radiograph of the spine [22].

### 2.4. Statistical Analysis

The data are presented as mean ± standard deviation, and the values that were not normally distributed are represented as medians (IQR). All statistical significance was set at a *p*-value < 0.05. Continuous variables were analyzed using Student’s *t*-test or Mann–Whitney test, and categorical variables were analyzed using the chi-square test or Fisher’s exact test. Logistic regression was used to explore osteoporosis-related variables according to the WHO criteria and high risk of fracture by the FRAX. FRAX with BMD, FRAX without BMD, and osteoporosis by the WHO criteria were used as the dependent variables. Variables based on statistical significance of univariate analysis (≤0.1 in univariate analysis) were used as independent variables and entered into a multivariable analysis using a stepwise method. All the variables except age and mSASSS (continuous variables) were binary variables in this model. The results of the regression analyses are expressed as odds ratios (ORs) with 95% confidence intervals (CIs). All statistical analyses were performed using the R software (version 4.0.1; R Foundation for Statistical Computing, Vienna, Austria).

## 3. Results

### 3.1. Baseline Characteristics of Patients with AS

A total of 219 patients with AS were enrolled in this study. FRAX was used for patients over 40 years of age, and only for 154 patients. The baseline characteristics of the AS patients are presented in Table 1. The mean age of the patients was 47.6 ± 13.8 years, 144 (65.8%) were men, and 43 (57.3%) were postmenopausal women. The demographic composition of the AS patients was not different from that of the HCs. The mean disease duration was 48.6 ± 46.6 months, and 183 (83.6%) patients were HLA-B27-positive. The clinical symptoms were peripheral arthritis in 84 (38.4%) patients, enthesopathy in 25 (11.4%) patients, and uveitis in 34 (15.5%) patients. The mean mSASSS at baseline was 20.7 ± 20.6, and 88 (40.2%) patients had syndesmophytes. Of the 219 patients, 25 (11.4%) had fractures, 20 (9.1%) had vertebral fractures, and 5 (2.3%) had non-vertebral fractures. Among the study patients, 33 (15.1%) were treated for osteoporosis; bisphosphonates were the most commonly used drug, followed by selective estrogen receptor modulators and denosumab. There were 206 (47.9%), 89 (40.6%), and 51 (23.3%) patients who received vitamin D, calcium, and proton pump inhibitors (PPI), which are medications affecting bone metabolism.

### 3.2. Evaluation of Osteoporosis by the WHO Criteria and of the Osteoporotic Fracture Risk by the FRAX and Comparison between Patients with AS and Healthy Controls

We compared BMD by DXA and the 10-year probability of osteoporotic fractures based on the FRAX calculation between the AS patients and the HCs, as shown in Figure 1. The distribution of the BMD categories in the AS patients was significantly different compared with that in the HCs. According to the WHO criteria, the percentage of normal BMD was 32.4% in the AS patients and 78.5% in the HCs. In the AS patients and HCs, the prevalence of osteopenia was 47% and 19.2%, respectively (*p* < 0.001), and that of osteoporosis was 20.1% and 2.3%, respectively (*p* < 0.001). The group at high risk for HF corresponded to 13.2% for AS patients and to 2.6% for HCs according to the FRAX with BMD (*p* = 0.001), and to 15.1% for AS patients and 5.3% for HCs according to FRAX without BMD (*p* = 0.007). There were also significant differences in the groups at high risk for MOF using the FRAX without BMD, which corresponded to 4.6% and 0% for AS patients and HCs, respectively (*p* = 0.02).

### 3.3. Treatment Status of Patients with a High Risk of Osteoporotic Fracture as Determined by the FRAX and of Osteoporosis According to the WHO Criteria

Patients at a high risk of osteoporotic fracture based on the FRAX calculation and on osteoporosis evaluation according to the WHO criteria require medication for osteoporosis treatment. The number of patients who were candidates for pharmacological treatment for fracture prevention is presented in Table 2. Based on the results of the FRAX calculation, the number of candidates for pharmacological treatment was significantly lower than the number of those selected according to the WHO osteoporosis criteria (FRAX with BMD vs. WHO osteoporosis criteria = 20 [13.2%] vs. 44 [20.1%], *p* < 0.001 and FRAX without BMD vs. WHO osteoporosis criteria = 23 [15.1%] vs. 44 [20.1%], *p* = 0.01). When using FRAX with BMD and FRAX without BMD, the number of candidates for pharmacological treatment was similar (FRAX with BMD vs. FRAX without BMD = 20 [13.2%] vs. 23 [15.1%], *p* = 0.678). 

Table 3 shows the treatment status of the patients at high risk of osteoporotic fractures by the FRAX calculation and osteoporosis evaluation according to the WHO criteria. According to the WHO criteria, 75% of patients with osteoporosis received pharmacological treatment, which was the highest percentage among the three groups. Patients at high risk of osteoporotic fractures according to the FRAX had significantly lower treatment rates than those determined according to the WHO criteria, corresponding to 50% for the FRAX with BMD and 39.1% for the FRAX without BMD. In all three groups, the proportion of patients receiving pharmacological treatment was higher for women than for men.

### 3.4. Clinical Factors Related to a High Risk of Fracture Calculated Using FRAX and Osteoporosis Evaluation Based on the WHO Criteria

The results of univariate analysis and multivariate analysis are shown in Appendix A and Table 4, respectively. In multivariate logistic regression, postmenopausal status was a consistent factor affecting all three assessments, based on FRAX with BMD (OR 4.36, CI 1.68–13.5, *p* = 0.005), FRAX without BMD (OR 5.7, CI 2.3–20.5, *p* < 0.001), and WHO osteoporosis criteria (OR 10.2, CI 1.53–196, *p* = 0.025). With FRAX without BMD, erythrocyte sedimentation rate (ESR) (OR 2.8, CI 0.74–5.32, *p* = 0.045), C-reactive protein (CRP) level (OR 3.78, CI 0.29–93, *p* 0.028), use of glucocorticoids (OR 1.54, CI 1.46–7.5, *p* = 0.02), and use of PPI (OR 1.68, CI 1.07–4.71, *p* = 0.035) have been proven to be factors that increase the risk of osteoporotic fractures. Human leukocyte antigen-B27 (HLA-B27) positivity (OR 5.3, CI 1.07–15.3, *p* = 0.046), CRP level (OR 1.67, CI 0.13–0.96, *p* = 0.041), and use of glucocorticoids (OR 6.02, CI 1.36–26.6, *p* = 0.018) were associated with osteoporosis based on the WHO criteria.

### 3.5. Analysis of Risk Factors Affecting Osteoporotic Fractures in Patients with AS

Of the 219 patients enrolled in this study, 25 had fractures. To analyze the risk factors associated with osteoporotic fractures, univariate and multivariate analysis were performed in patients with AS (Table 5). A FRAX with BMD score (OR 8.56, CI 2.32–31.5, *p* = 0.001) and an mSASSS for fracture (OR 1.03, CI 0.96–1.01, *p* = 0.043) were identified as risk factors that significantly affected osteoporotic fracture. To compensate for the false elevation of the FRAX score due to the history of previous fracture, a modified FRAX score without this factor was calculated. As a result, only the modified FRAX score without BMD showed a significant difference (Appendix A). In our analysis, BMD measurements using DXA did not act as a risk factor for osteoporotic fracture (*p* = 0.99).

## 4. Discussion

The reported prevalence of osteoporosis in AS is wide, ranging from 18.7% to 62%, as BMD may decrease in the early stages of AS and may increase with the formation of syndesmophyte in the later stages of the disease [11,12,13,14,23]. In this comparative analysis, the BMD of the spine and femur was lower in AS patients than in age- and sex-matched Korean HCs, similar to previous results [23,24]. In the FRAX calculations, the proportion of patients at a high risk of HF among AS patients was significantly higher than that among HCs. Although the AS patients had a higher probability of osteoporotic fractures based on the FRAX calculation than the HCs, it was not higher than the probability of other joint diseases [25,26,27]. Previous studies used the FRAX calculation to evaluate the risk of osteoporotic fractures and showed that 47.2–61% of patients with RA and 19.1–38.1% of patients with osteoarthritis (OA) were at high risk of osteoporotic fractures. Considering that patients with RA and OA had a longer disease duration and old age compared to the patients in this study, there must have been differences between the diseases, considering the absolute value of FRAX. Several studies on patients with AS showed that the risk of MOF and HF was 1–17%, similar to our results [28,29,30]. Given the parameters used in the FRAX calculation, it is estimated that the proportion of patients with AS at high risk of osteoporotic fractures is relatively low. AS begins at a young age and requires a more limited glucocorticoid treatment than other chronic inflammatory diseases [19]. Furthermore, FRAX calculation for patients with AS could be relatively underestimated, as RA is the only disease considered a secondary factor based on the FRAX tool.

In the analysis of pharmacological candidates for fracture prevention by comparing WHO criteria and FRAX calculation, the number of candidates meeting the WHO criteria was significantly higher than that obtained using the FRAX calculation. As mentioned above, these results are presumed to be due to patients’ characteristics such as age and sex, as well as to therapeutic agents administered to the patients with AS. This tendency was also observed for SLE, whereas RA and OA had a higher proportion of patients at high risk of osteoporotic fractures according to FRAX compared to that determined using the osteoporosis evaluation by the WHO criteria [25,26,27]. Unfortunately, only 39.1–75% of the patients selected as candidates for pharmacological therapy received treatments for fracture prevention. The actual treatment rate was significantly lower than necessary, especially on the basis of the FRAX calculation than the basis of the WHO criteria, and for men compared to women. This is probably because the insurance requirements for osteoporosis treatment in Korea are based on BMD according to the WHO criteria. While most rheumatic diseases have a high incidence in women, AS is predominant among men; thus, evaluation and treatment for fracture prevention in men should be emphasized as part of the treatment course for AS. However, pharmacological treatment is challenging for premenopausal women or men before the age of 50 years because of a lack of robust evidence of its benefits, and there are also limitations on available medications [31]. For these reasons, the majority of premenopausal women and young men with osteoporosis may have been treated with lifestyle changes (e.g., quit alcohol consumption and smoking) or alternative medications such as calcium or vitamin D supplementation [32].

The exact mechanisms and causes of bone loss and fractures in AS have not yet been fully elucidated. Various factors have been suggested as possible explanations, including systemic inflammation, decreased physical activity, use of non-steroidal anti-inflammatory drugs, and smoking [33,34,35,36]. In our data, menopause, ESR, CRP level, HLA-B27 positivity, glucocorticoid use, and PPI use were associated with an increased prevalence of osteoporosis and a high risk of osteoporotic fractures. Among these factors, menopause was the most influential factor involved in both criteria. However, it is worth noting that factors other than menopause play a role, given that menopause is already widely known as the main cause of bone loss due to declining estrogen levels [37]. In detail, ESR, CRP level, glucocorticoid use, and PPI use appeared to be related to a high risk of osteoporotic fractures based on the FRAX without BMD. ESR and CRP level are objective measurements representing inflammation and may vary depending on the activity of AS. Our results suggest that uncontrolled disease activity in AS leads to the risk of osteoporotic fractures, which is consistent with the findings of some studies [38] but disagrees with others [39,40]. Several studies have also shown a negative correlation between the Bath Ankylosing Spondylitis Disease Activity Index (BASDAI), the Bath Ankylosing Spondylitis Functional Index (BASFI), and BMD [41,42]. In addition, the association of the use of glucocorticoids and PPIs with a high risk of osteoporotic fractures, as shown in this study, is in line with previous findings that the risk of fractures increases by known mechanisms such as increased bone resorption, decreased bone formation, and ultimately reduced BMD. Although the mechanism of PPIs is unclear, they may affect bone density by reducing calcium absorption or by inducing hyperparathyroidism due to hypergastrinemia [43,44,45]. On the other hand, age is usually the strongest predictor of bone loss and fractures; however, age was not a significant factor in multivariate analysis in our study, which can be explained by the increased in BMD due to syndesmophyte as the disease progressed.

HLA-B27 positivity was related to osteoporosis based on the WHO criteria, but this was not seen when using the FRAX calculation. A worldwide AS registry study of the relationship between the presence of HLA-B27 and the comorbidities associated with AS also noted a higher frequency of osteoporosis in HLA-B27-positive patients than in HLA-B27-negative patients [46]. The role of HLA-B27 in bone metabolism has not yet been studied in humans; however, experimental studies conducted on HLA-B27-positive transgenic rats have shown that bone loss occurs through the intervention of the receptor activator of nuclear factor-κB ligand/osteoprotegerin systems or the alteration of bone material properties [47,48]. The potential relationship between HLA-B27 and bone metabolism in humans is expected to be similar to that in animals, and more appropriate studies should be performed in these genotype-positive patients.

Our analysis found a general similarity of clinical factors leading to a high risk of osteoporotic fracture according to FRAX and osteoporosis evaluation according to the WHO criteria. Decisions on treatments for fracture prevention suggest the need to aggregate clinical aspects as well as BMD values. FRAX is not a more reliable and effective evaluation tool in deciding whether to treat patients with fracture prevention medication; notwithstanding, it is problematic to adopt treatment targets only based on the BMD, currently used primarily [49]. The National Bone Health Alliance Working Group has proposed continuing the use of T-scores as a means of diagnosing osteoporosis but has expressed the position that a high-risk of osteoporotic fractures also confers the diagnosis of osteoporosis [50]. Even if AS occurs mostly in young men, applying the WHO and FRAX criteria in combination will certainly help prevent osteoporotic fractures. The need for FRAX calculation is emphasized, especially for elderly patients with long disease duration, as the accuracy of BMD values measured with DXA decreases. In addition, since BMD can be inaccurate in the evaluation of the osteoporosis treatment response, FRAX calculation or bone turnover markers measurement can be a good alternative as an evaluation tool.

Among the patients with AS included in the study, the mSASSS and proportion of patients at high risk of osteoporotic fracture according to the FRAX calculation showed statistically significant differences between patients with and without osteoporotic fractures. In the FRAX calculation, the history of previous fractures is input as a risk factor; therefore, the high probability of fracture based on FRAX calculation in the fracture group is a predictable outcome. The modified FRAX without the history of previous fracture was calculated to correct this, and the results were similar (Appendix A). In other words, we not only provided evidence that FRAX calculation has clinical value as a fracture risk prediction tool but also support the need to consider the FRAX calculation in decision making related to osteoporosis treatment. Another significant factor, the mSASSS, showed a positive association with fracture occurrence, consistent with several previous findings [51].

The strength of this analysis is that, to the best of our knowledge, this is the first multicenter study conducted to estimate the risk of osteoporotic fractures by comparing WHO and FRAX criteria in patients with AS. However, our study has some limitations. First, the current study only analyzed data available from retrospective studies; thus disease activity assessment could be incomprehensive. As an evaluation method for disease activity in AS, BASDAI, BASFI, visual analog scale, and patient and physician global assessments should have been used during the clinic visits. We only obtained data on inflammatory markers such as ESR and CRP. Further prospective studies are needed to establish a definitive relationship between various disease activity indices and osteoporotic fracture risk in AS. Second, selection bias occurred during data collection owing to the characteristics of the study design because the study included patients with AS who had a BMD test. Finally, factors that can affect bone metabolism, such as osteoporosis therapy and vitamin D levels, were not fully compensated.

## 5. Conclusions

Among 219 patients with AS in this study, 44 (20.1%), 20 (13.2%), and 23 (15.1%) patients required pharmacological treatment according to the WHO criteria, FRAX with BMD, and FRAX without BMD, respectively; of these, only 39.1–75% received actual treatment. For the evaluation of the risk of fractures in patients with AS, it would be helpful to consider both the FRAX tool with and without BMD and the WHO criteria, especially for patients with elevated inflammatory markers or taking glucocorticoids or PPIs.

## Figures and Tables

**Figure 1 jcm-11-02830-f001:**
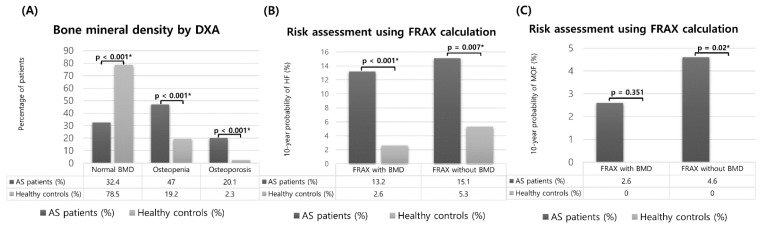
Comparison of bone mineral density (BMD) scores and 10-year fracture probabilities based on fracture risk assessment tool (FRAX) calculations between patients with ankylosing spondylitis and healthy controls after age, sex, and body mass index matching. (**A**) Bone mineral density by dual energy x-ray absorptiometry. (**B**,**C**) Risk assessment using FRAX calculation. * *p* < 0.05.

**Table 1 jcm-11-02830-t001:** Baseline characteristics of the patients with ankylosing spondylitis.

Variable	AS Patients (*n* = 219)	Healthy Controls (*n* = 219)	*p*-Value
** Demographics **			
Age, mean (SD), years	47.6 (13.8)	47.5 ± 13.7	0.972
Sex, N (%)			
Men	144 (65.8)	144 (65.8)	
Women	75 (34.2)	75 (34.2)	
Menopause, N (%)	43 (57.3)	40 (53.3)	0.628
Bodyweight, mean (SD), kg	64.3 (12.2)	66.4 (11.3)	0.71
Height, mean (SD), cm	163.9 (10.1)	165.6 (8.41)	0.06
Body mass index, mean (SD), kg/m^2^	23.9 (3.7)	24.3 (3.09)	0.227
Smoking, N (%)	53 (24.2)	41 (18.7)	0.082
Alcohol, N (%)	53 (24.2)	68 (31.1)	0.225
Fractures, N (%)	25 (11.4)	3 (1.7)	<0.001 *
Vertebral fractures, N (%)	20 (9.1)	3 (1.7)	<0.001 *
Non-vertebral fractures, N (%)	5 (2.3)	0 (0)	0.025 *
** Disease-related **			
Disease duration, mean (SD), months	48.6 (46.6)		
HLA-B27-positive, N (%)	183 (83.6)		
ESR, mm/h, median (IQR)	31 (13–59)		
CRP, mg/dL, median (IQR)	1.57 (0.39–3.02)		
Peripheral arthritis, N (%)	84 (38.4)		
Enthesopathy, N (%)	25 (11.4)		
Uveitis, N (%)	34 (15.5)		
Syndesmophyte, N (%)	88 (40.2)		
mSASSS score, median (IQR)	13 (5–32)		
** Medications **			
NSAIDs, N (%)	190 (86.8)		
Methotrexate, N (%)	59 (26.9)		
Sulfasalazine, N (%)	137 (62.6)		
TNF inhibitor, N (%)	98 (44.7)		
GC, N (%)	69 (31.5)		
GC lifetime use, mean (SD), g (prednisone-equivalent dose)	10.7 (2.04)		
GC current dose, mean (SD), mg/d	1.69 (3.13)		
Vitamin D intake, N (%)	105 (47.9)		
Calcium intake, N (%)	89 (40.6)		
Proton pump inhibitor, N (%)	51 (23.3)		
Treatment for osteoporosis, N (%)	33 (15.1)		
Bisphosphonate, N (%)	24 (11)		
SERM, N (%)	6 (2.7)		
Denosumab, N (%)	3 (1.4)		

AS, ankylosing spondylitis; HLA, human leukocyte antigen; ESR, erythrocyte sedimentation rate; IQR, interquartile range; CRP, C-reactive protein; mSASSS, modified Stoke Ankylosing Spondylitis Score; NSAIDs, non-steroidal anti-inflammatory drugs; TNF, tumor necrosis factor; GC, glucocorticoid; SERM, selective estrogen receptor modulators; * *p* < 0.05.

**Table 2 jcm-11-02830-t002:** Distribution of patients at high risk of fracture based on the FRAX with and without BMD and on osteoporosis evaluation according to the WHO criteria among patients with ankylosing spondylitis.

	High Risk of Fracture in FRAX with BMD	High Risk of Fracture in FRAX without BMD	Osteoporosis by the WHO Criteria	*p* Value ^1^	*p* Value ^2^	*p* Value ^3^
Overall	20/152 (13.2%)	23/152 (15.1%)	44/219 (20.1%)	0.678	<0.001 *	0.01 *
Men	11/88 (12.5%)	6/88 (6.8%)	27/144 (18.8%)	<0.001 *	0.006 *	<0.001 *
Women	9/64 (14.1%)	17/64 (26.6%)	17/75 (22.7%)	<0.001 *	0.029 *	0.007 *

FRAX, Fracture Risk Assessment Tool (%, 10-year probability of major osteoporotic and vertebral fracture, respectively, corresponding to patients aged ≥40 years); BMD, bone mineral density; WHO, World Health Organization. * *p* < 0.05. ^1^ FRAX with BMD vs. FRAX without BMD, ^2^ FRAX with BMD vs. WHO criteria, ^3^ FRAX without BMD vs. WHO criteria.

**Table 3 jcm-11-02830-t003:** Current osteoporosis treatments in patients at high risk of fracture based on the FRAX and on osteoporosis evaluated according to the WHO criteria.

	FRAX with BMD	FRAX without BMD	Osteoporosis of the WHO
Candidates for pharmacological treatment	20/152 (13.2%)	23/152 (15.1%)	44/219 (20.1%)
** Current treatment **			
Overall	10/20 (50%)	9/23 (39.1%)	33/44 (75%)
Men	5/11 (45.5%)	2/6 (33.3%)	18/27 (66.7%)
Women	5/9 (55.6%)	7/17 (41.2%)	15/17 (88.2%)

FRAX, Fracture Risk Assessment Tool (%, 10-year probability of major osteoporotic and vertebral fracture, respectively, corresponding to patients aged ≥40 years); BMD, bone mineral density; WHO, World Health Organization.

**Table 4 jcm-11-02830-t004:** Variables associated with a high risk of fracture based on the FRAX with and without BMD and osteoporosis evaluation according to the WHO criteria.

	FRAX with BMD		FRAX without BMD		Osteoporosis (WHO)	
	OR (95% CI)	*p*-Value	OR (95% CI)	*p*-Value	OR (95% CI)	*p*-Value
Age	1.12 (0.99, 1.26)	0.062	1.94 (0.98, 3.86)	0.058	1.05 (0.98, 1.12)	0.156
Sex (Men)	0.64 (0.28, 1.47)	0.293	0.87 (0.79, 3.1)	0.102	0.31 (0.01, 1.66)	0.117
Menopause	4.36 (1.68, 13.5)	0.005 *	5.66 (2.3, 20.5)	<0.001 *	10.2 (1.53, 196)	0.025 *
BMI < 25 kg/m^2^	0.99 (0.09, 3.89)	0.59	0.22 (0.02, 2.75)	0.24	2.4 (0.62, 9.21)	0.203
HLA-B27 positivity	4.22 (0.81, 77.8)	0.171	2.89 (0.74, 5.32)	0.089	5.3 (1.07, 15.3)	0.046 *
ESR	1.2 (0.46, 27.5)	0.139	2.78 (1.01, 4.68)	0.045 *	1.27 (0.13, 1.81)	0.278
CRP	1.25 (0.89, 1.78)	0.203	3.78 (0.29, 93)	0.028 *	1.67 (0.13, 0.96)	0.041 *
Syndesmophyte	0.22 (0.21, 2.35)	0.210	1.28 (0.17, 4.84)	0.465	1.09 (0.25, 4.63)	0.913
mSASSS	0.97 (0.9, 1.04)	0.344	1.14 (0.96, 1.36)	0.136	0.98 (0.94, 1.02)	0.245
Glucocorticoid use	4.32 (0.59, 31.4)	0.149	1.54 (1.46, 7.5)	0.02 *	6.02 (1.36, 26.6)	0.018 *
Biologics use	1.01 (0.99, 1.12)	0.059	1.34 (0.55, 3.32)	0.515	1.43 (0.33, 6.18)	0.09
PPI use	1.3 (0.46, 3.41)	0.603	1.68 (1.07, 4.71)	0.048 *	0.35 (0.08, 1.55)	0.167
Vitamin D use	0.61 (0.05, 7.17)	0.692	0.3 (0.01,1.02)	0.051	0.45 (0.06, 3.5)	0.447
Calcium use	1 (0.12, 8.56)	0.997	0.07 (0, 17.5)	0.349	0.62 (0.1, 3.78)	0.604

FRAX: Fracture Risk Assessment Tool (%, 10-year probability of major osteoporotic and vertebral fracture, respectively, corresponding to patients aged ≥40 years); BMD: bone mineral density; WHO: World Health Organization; OR: odds ratio; CI: confidence interval; BMI: body mass index; HLA: human leukocyte antigen; ESR: erythrocyte sedimentation rate; CRP: C-reactive protein; mSASSS: modified Stoke Ankylosing Spondylitis Score; PPI: proton pump inhibitor. * *p* < 0.05.

**Table 5 jcm-11-02830-t005:** Univariate and multivariate analysis of risk factors associated with osteoporotic fractures.

	Osteoporotic Fracture			
	Univariable Model		Multivariable Model	
	OR (95% CI)	*p*-Value	OR (95% CI)	*p*-Value
Age	1.04 (1.01, 1.07)	** 0.015 **	0.98 (0.91, 1.06)	0.683
Sex (Men)	0.92 (0.38, 2.19)	0.844		
Menopause	1.19 (0.44, 3.23)	0.734		
Disease duration	0.99 (0.98, 1.00)	** 0.097 **	0.98 (0.96, 1.01)	0.12
BMI	1.03 (0.92, 1.15)	0.648		
HLA-B27 positivity	1.42 (0.31, 6.47)	0.652		
ESR	1.01 (0.99, 1.02)	0.862		
CRP	0.98 (0.85, 1.14)	0.822		
Syndesmophyte	2.03 (0.85, 4.86)	0.113		
mSASSS	1.03 (1.01, 1.05)	** 0.007 **	1.03 (1.01, 1.06)	0.043 *
Glucocorticoid cumulative dose	1.05 (0.88, 1.26)	0.571		
Biologics use	1.14 (0.49, 2.61)	0.766		
PPI use	1.9 (0.77, 4.66)	0.164		
FRAX with BMD	6.35 (2.07, 19.5)	** 0.001 **	8.56 (2.32, 31.5)	0.001 *
FRAX without BMD	6.19 (2.11, 18.1)	** 0.001 **	2.67 (0.47, 15.3)	0.271
BMD by DXA	1.01 (0.36, 2.85)	0.99		

BMI, body mass index; HLA, human leukocyte antigen; ESR, erythrocyte sedimentation rate; CRP, C-reactive protein; mSASSS, modified Stoke Ankylosing Spondylitis Score; PPI, proton pump inhibitor; FRAX, Fracture Risk Assessment Tool (%, 10-year probability of major osteoporotic and vertebral fracture, respectively, corresponding to patients aged ≥40 years); BMD, bone mineral density; DXA, dual-energy X-ray absorptiometry. Bold statistics denote *p*-value ≤ 0.1 and * *p* < 0.05.

## Data Availability

The datasets generated for this study are not available due to the data protection law.

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
