# Peer review of "Prevalence and Factors of Osteoporosis and High Risk of Osteoporotic Fracture in Patients with Ankylosing Spondylitis: A Multicenter Comparative Study of Bone Mineral Density and the Fracture Risk Assessment Tool"

_jcm, 2022, doi:10.3390/jcm11102830_

Round 1

Reviewer 1 Report

This is an interesting article that emphasizes the need for osteoporosis surveillance in AS patients as compared to healthy controls. Patient demographics, laboratory data, and BMD measured by DXA were retrospectively compared to presence of a high-risk FRAX score and diagnosis made by the WHO criteria, with the finding that AS patients were at significantly higher risk of osteoporosis. The authors also found that more patients met the threshold for treatment using the WHO criteria compared to those who had high-risk FRAX scores warranting treatment. The observation that menopause was significantly associated with high-risk FRAX score and WHO criteria can be taken as convergent validity in support of the results of this study. Markers of inflammation were also found to be correlated with WHO criteria in addition to the FRAX score calculated without BMD. Ultimately, both FRAX and WHO criteria were found to be relevant to osteoporosis management, which is reasonable based on their findings. As a retrospective study with some limitations, the results are nonetheless relevant to clinical practice, adding nuance to the use of clinical tools and highlighting the importance of osteoporosis screening and treatment in both men and women with AS.

Minor Comments:

1. ln 71-72, 359-360: Limitations of DXA (especially spinal DXA) are mentioned in the introduction and conclusion. It may be worth adding a sentence describing the expected role of bone turnover markers in osteoporosis management.

2. ln 301-305: The discrepancy in patients who meet threshold for treatment based on FRAX vs WHO criteria is described. Please mention the suspected reason for this discrepancy.

3. ln 398-400: The conclusion states that both the FRAX and WHO criteria should be used for the assessment of osteoporosis in AS patients. Please clarify if this applies to FRAX with BMD, FRAX without BMD, or both.

Author Response

Thank you for your valuable comment.

Minor Comments:

  1. ln 71-72, 359-360: Limitations of DXA (especially spinal DXA) are mentioned in the introduction and conclusion. It may be worth adding a sentence describing the expected role of bone turnover markers in osteoporosis management.

Answer> Thank you for your comment. As your comment, we added the contents to the discussion section as follows and underlined it in the text.

  • In addition, since BMD can be inaccurate in the evaluation of osteoporosis treatment response, FRAX calculation or bone turnover marker measurement can be a good alternative as an evaluation tool.
  1. ln 301-305: The discrepancy in patients who meet threshold for treatment based on FRAX vs WHO criteria is described. Please mention the suspected reason for this discrepancy.

Answer> Thank you for your comment. We revised the text as follows.

  • In the analysis of pharmacological candidates for fracture prevention by comparing WHO criteria and FRAX calculation, the number of candidates meeting the WHO criteria was significantly higher than that using the FRAX calculation. As mentioned above, these results are presumed to be due to disease characteristics such as age, sex, and therapeutic agents of patients with AS. This tendency was consistent with SLE, whereas RA and OA had a higher proportion of high-risk of osteoporotic fractures in FRAX compared to osteoporosis by WHO criteria
  1. ln 398-400: The conclusion states that both the FRAX and WHO criteria should be used for the assessment of osteoporosis in AS patients. Please clarify if this applies to FRAX with BMD, FRAX without BMD, or both.

Answer> Thank you for your comment. We recommended using both FRAX with BMD and without BMD, so we added it as follows.

  • For the evaluation of the risk of fractures in patients with AS, it would be helpful to consider both FRAX tool with and without BMD and WHO criteria, especially in patients with elevated inflammatory markers or taking glucocorticoids or PPI.

Reviewer 2 Report

In the present study, authors aimed to identify, among patients with ankylosing spondylitis (AS), those with osteoporosis using the WHO criteria and those with a high-risk of osteoporotic fractures according to FRAX. The proportion of patients receiving treatment and the factors related to osteoporotic fractures were also identified.
To do so, they retrospectively reviewed the records of patients with AS from January 2012 to June 2020 at five university hospitals in South Korea. They paired each AS patient with a healthy control.
Of the 219 patients with AS, 25 (11.4%) had fractures, 20 (9.1%) had vertebral fractures, and 5 (2.3%) had non-vertebral fractures. The authors do not report the frequency of those fractures among healthy controls. I suggest retrieving and presenting that information.
Given the retrospective nature of the study, it would be interesting to extend the data retrieval period backwards in order to confirm that the estimation for the incidence of fractures at 10 years obtained by FRAX have been fulfilled or not in the following 10 years in patients with AS. Also, it would of interest for the clinician to know whether the precision of those estimates differs between AS patients and healthy controls.
I suggest performing multivariate analysis of the risk factors associated with the appearance of osteoporotic fractures in patients with AS.
As a conclusion, authors encourage to consider both FRAX tool and WHO criteria in patients with AS, especially in those patients with elevated inflammatory markers or taking glucocorticoids or PPI. I suggest the authors to build, test and validate a new score that would take into account all the evidence collected in the present paper. That score could be validated using a different cohort of patients suffering from AS, collected either retrospective or prospectively.

Author Response

We attached file.

Reviewer 3 Report

The manuscript titled, "Prevalence and Factors of Osteoporosis and High-Risk of Osteoporotic Fracture in Patients with Ankylosing Spondylitis: A Multicenter Comparative Study of the Bone Mineral Density and the Fracture Risk Assessment Tool" explored the prevalence and associated factors of high-risk osteoporotic fractures in Korean patients with ankylosing spondylitis (AS). The results of this important study suggest that both FRAX and WHO criteria may be complementary to treatment decisions to reduce osteoporotic fractures in patients with AS. The manuscript is well written and the results reported in the study support the conclusions made in the study.

Author Response

Thank you for your valuable comment.

Round 2

Reviewer 2 Report

Dear authors,

Thank you for answering my questions.  The paper is more relevant now with the changes made. 

I highly recommend to extend it as suggested previously in further publications. 

Good luck